# Human biomass movement exceeds the biomass movement of all land animals combined

Yuval Rosenberg ®[1], Dominik Wiedenhofer ®[2], Doris Virág[2], Gabriel Bar-Sella[1], Lior Greenspoon ®[1], Barr Herrnstadt[1], Lewis Akenji ®[3], Rob Phillips[4,5] & Ron Milo ®[1]✉

Earth is teeming with life on the move, shaping ecosystems and human civilizations alike. However, the magnitude of movement by humans and other animals has yet to be assessed holistically. Here we quantify the movement of biomass across all animal life and in comparison to humanity. We show that the combined biomass movement of all wild birds, land arthropods and wild land mammals is about one-sixth that of humans walking and about 40 times smaller than all the biomass movement of humans. The biomass movement of marine animals, which we find to be the living world's largest, has been halved since 1850 due to industrial fishing and whaling, while human biomass movement has increased by about 40-fold. This study gives a quantitative perspective on global mobility in the Anthropocene and sharpens our perception regarding the extent of human versus animal activity.

Mobility is central and common to wild animals and humans alike[1]. It is a defining feature of animals, sometimes travelling thousands of kilometres each year while actively migrating[2], foraging, searching for mates and so on. Mobility is also essential to the daily lives of humans and their participation in society. As animals and humans move, they shape ecosystems in myriad ways[3,4], from transporting nutrients and organisms to trophic effects and physical ecosystem engineering. Mobility can thus serve as a concrete and direct comparison between humans and animals, which is consequential and intuitive.

This led us to ask: how does the total mobility of humans compare with that of all wild animals combined? To our knowledge, human versus animal mobility has yet to be addressed comparatively or systematically on a global scale[1]. Here, we define the biomass movement of a given species as its total biomass times the distance it actively travels per year (having units of annual biomass-distance, that is, units of mass times speed, like those of momentum). This metric makes it possible to compare human and animal mobility[5] and extends the passenger-distance units commonly used to unify and analyse modes of human transportation.

We synthesized hundreds of studies and data sources and used diverse approaches to coherently evaluate the biomass movement of all animals and humans on Earth. We recognize that future high-resolution monitoring might increase current estimates[6]. Only a fraction of species was monitored for travelling distances. However, they represent characteristic movement patterns of central animal groups and the total biomass of many species is small, allowing us to make order-of-magnitude estimates of global biomass movement. We include different modes of locomotion, taxonomy and timescales. As exemplified below, the biomass movement metric offers a valuable perspective on humanity as part of the biosphere.

## Results

We quantified the active biomass movement of a taxonomic group as the product of their biomass and the distance they actively travel per unit of time. We estimated the biomass stock and typical movement patterns of all major groups of organisms. We grouped them on the basis of their mode of locomotion, taxonomy, movement-related traits and data availability. For all modes of human transportation,

[1]Department of Plant and Environmental Sciences, Weizmann Institute of Science, Rehovot, Israel. [2]Institute of Social Ecology, BOKU University, Vienna, Austria. [3]Hot or Cool Institute, Berlin, Germany. [4]Department of Physics, California Institute of Technology, Pasadena, CA, USA. [5]Division of Biology and Biological Engineering, California Institute of Technology, Pasadena, CA, USA. ✉e-mail: ron.milo@weizmann.ac.il

we included only human biomass. The mass movement of vehicles (excluding human biomass) is estimated and discussed separately. We provide our main results below, while Supplementary Information fully describes our estimates. The section on 'Sensitivity and uncertainty analysis' in Supplementary Information explains how we treat uncertainties and provides detailed uncertainty estimates for every animal group considered in the paper.

## Land animals

Figure 1 shows the main groups that contribute to biomass movement on land. The biomass movement of each such group is calculated as the sum of the estimated biomass movement of all the species in that group, as described in Supplementary Information. For example, we estimated the combined biomass movement of all wild land mammals, excluding bats, to be 30 Gt km yr$^{-1}$ (uncertainty range 10–70 Gt km yr$^{-1}$). We divide the ~30 Gt km yr$^{-1}$ total biomass movement by their combined biomass of 20 Mt (ref. 7) (uncertainty range 13–38) to find a biomass-weighted average daily distance of 4 km d$^{-1}$ (uncertainty range 2–5 km d$^{-1}$) (Fig. 1a). We also estimated an upper bound of ~150 Gt km yr$^{-1}$, accounting for possible systematic biases (Supplementary Information). Figure 1b shows the biomass movement estimates with uncertainty ranges and upper bounds for all wild terrestrial animals, humans and livestock. The uncertainty ranges for vertebrates and humans are based on statistical analyses (Supplementary Information). Land arthropod uncertainty range is based on minimal and maximal values and biological or ecological constraints instead of 95% confidence intervals (CIs) which are difficult to derive owing to the scarcity of data (Supplementary Information). For animal groups without sufficient data only upper-bound estimates were evaluated, having a combined biomass movement of less than 100 Gt km yr$^{-1}$.

We found that large animals dominate the biomass movement of land mammals. Large-bodied mammals need to forage extensively and they move with a low energetic cost of transport (COT)[8] (see below). Mammals whose adult mass is over 50 kg contribute ~50% of the biomass of wild land mammals, excluding bats, but ~80% of their total biomass movement. The African savannah elephant alone contributes ~23% of their biomass movement (while representing ~6% of their global biomass).

At higher taxonomic levels, small invertebrates completely dominate wild animal biomass on land, but most move much less than larger vertebrates, resulting in small overall biomass movement. All birds and all arthropods have a similar combined biomass movement despite birds having a total biomass of only ~3 Mt or ~300 times less biomass than terrestrial arthropods (Fig. 1). We estimate an upper bound of ~130 Gt km yr$^{-1}$ for wild birds and ~300 Gt km yr$^{-1}$ for terrestrial arthropods, accounting for possible biases.

In comparison, we found that the biomass movement of people is 4,000 Gt km yr$^{-1}$ (uncertainty range 3,400–7,000 Gt km yr$^{-1}$) (Fig. 1), over 40 times greater than our best estimate for all wild land mammals, arthropods and birds combined and over six times greater than the upper estimate for the biomass movement of all land animals combined. With about 8 billion people and an average weight of ~54 kg per person (Supplementary Information), the biomass of humanity is 0.43 ± 0.02 Gt—an order of magnitude higher than the biomass of all other wild terrestrial vertebrates. Humans also move longer average distances of ~30 km d$^{-1}$ (including via motorized transportation).

Most human biomass movement uses motorized vehicles, with ~65% in cars and motorcycles, ~10% in airplanes and ~5% in trains and subways. Two-thirds of all motorized mobility occurs in high-income and upper-middle-income countries[9]. However, walking still correspond to over 10% of human biomass movement (600 Gt km yr$^{-1}$ with uncertainty range 400–700 Gt km yr$^{-1}$), probably exceeding all terrestrial animals combined (~100 Gt km yr$^{-1}$ with an upper estimate of ~400 Gt km yr$^{-1}$ and an upper bound of ~700 Gt km yr$^{-1}$; Fig. 1c). In the air, the biomass movement of flying wild animals (~40 Gt km yr$^{-1}$ with

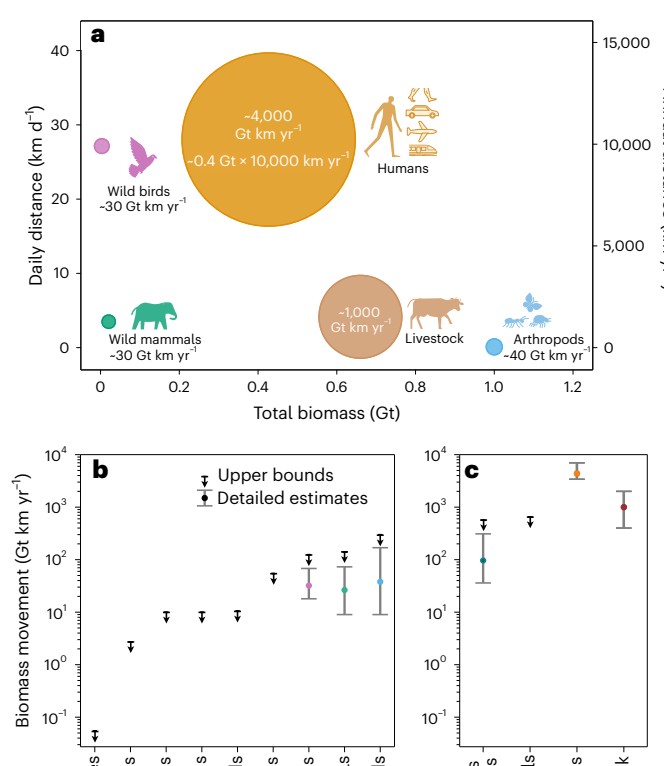

**Fig. 1 | Total biomass movement of land animals and birds. a**, The area of each circle is proportional to the total annual biomass movement of the corresponding group in Gt km yr$^{-1}$. Distances are biomass-weighted averages over all species in each group over a year. Wild mammals exclude bats. **b,c**, Mean biomass movement estimates (dots) with uncertainty ranges (error bars) and upper bounds (bars with downwards arrows) for all terrestrial animals, humans and livestock. **b**, Wild animal estimates by taxonomic group. **c**, Sums of the biomass movement of wild animal groups shown in **b**, compared with that of humans and livestock. Each estimate aggregates values from constituent subgroups such as different species or different modes of transportation. Error bars represent the arithmetic sum of the uncertainties of the subgroups, which are based on 95% CIs or extrema, to conservatively account for potential systematic biases. See Supplementary Information for more details on the data used for analysis, estimation methods, uncertainties and upper bounds.

an upper bound of less than 300 Gt km yr$^{-1}$) is much smaller than that of humans flying in airplanes (500 Gt km yr$^{-1}$ with uncertainty range 400–700 Gt km yr$^{-1}$). Domesticated animals have biomass movement of the same order of magnitude as humans, 1,000 ± 600 Gt km yr$^{-1}$. Locomotion of non-dairy cattle corresponds to most of this biomass movement.

## Prominent migrations and the dominance of marine biomass movement

The magnitudes of total biomass movement can be further appreciated by comparing prominent case studies, as summarized in Fig. 2. The spectrum of animal migrations spans many orders of magnitude of biomass movement. Flying animals may migrate great distances, but have low biomass movement due to their low total biomass. Roughly 2 million arctic terns (*Sterna paradisaea*) migrate annually from pole to pole, covering distances longer than any other animal. However, their body mass is only ~100 g, so their total biomass

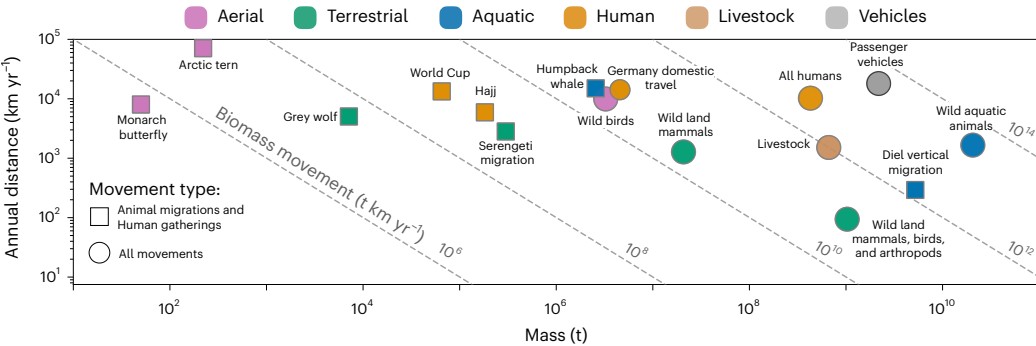

**Fig. 2 | Biomass movement across prominent migrations and gatherings, and total mass movement across groups.** Squares represent migrations, with distance summed over a year. Axes are on a logarithmic scale. Dashed lines depict lines of constant mass movement in units of t km yr⁻¹, representing constant multiplications between the mass (horizontal axis) and annual distance (vertical axis). Circles represent the total mass movement of entire groups, extending beyond specific migrations or gatherings. Circles include data presented in Figs. 1 and 3. See Supplementary Information for detailed estimates.

movement is only ~16 million t km yr⁻¹ (~0.016 Gt km yr⁻¹). The biomass movement of arctic terns is about half the global biomass movement of grey wolves (*Canis lupus*), which travel especially long distances for land mammals[10], with a biomass movement of ~0.03 Gt km yr⁻¹. The migration of over a million blue wildebeest (*Connochaetes taurinus*), gazelles and zebras of the Serengeti are an icon of ungulate mass migrations. Their annual biomass movement is ~20 times larger than that of grey wolves. Putting it in a human perspective, it is similar to the biomass movement associated with international human gatherings such as the Muslim Hajj (~2 million pilgrims) or the FIFA World Cup (~1 million spectators).

We find that the ocean is the location of vastly larger biomass movements. Humpback whales (*Megaptera novaeangliae*) of ~30 t of body mass travel from tropical breeding grounds to feeding grounds near the poles, with a global population of ~80,000 mature whales. Their long migration alone has a similar biomass movement to that of all land mammals or all birds combined. It is also similar to domestic travel in Germany, which includes ~80 million people with a similar total biomass and a travel distance of ~15,000 km yr⁻¹ (~40 km d⁻¹). While massive whales migrate impressive long distances, the daily vertical movement of zooplankton and mesopelagic fish has far greater biomass movement. They compose most of the animal biomass on Earth[11,12] and, every day, 15–50% of them swim up and down the water column to forage for food and escape predators[13]. The zooplankton and mesopelagic fish migrate ~1 km daily with a total biomass of ~5 Gt. This so-called diel vertical migration, which exceeds 4 Gt km d⁻¹, or ~1,000 Gt km yr⁻¹, surpasses any other animal migration in biomass movement. It has far greater biomass movement than all wild land animals combined and is on par with humans walking and cycling.

While the diel vertical migration appears to have maintained most of its original abundance[11], the ocean as a whole lost ~60% of its biomass movement since the year 1850, primarily because of commercial fishing and whaling, from ~80,000 Gt km yr⁻¹ to ~30,000 Gt km yr⁻¹ (Fig. 3). Humanity, therefore, decreased by more than 50% the total biomass movement in our oceans. The total biomass of fish and marine mammals in the upper ocean is now at least 40% (ref. 11) lower than in 1850. Large-bodied animals, which travel more, declined the most. During the same time, human biomass movement increased 40-fold. While human walking was responsible for the vast majority of human biomass movement in 1850, today, walking corresponds to roughly a seventh of human biomass movement. High-income countries[9] have seen the largest increase in per capita biomass movement, almost twice as much as in other income groups (Supplementary Fig. 1). They host 16% of the global population but ~30% of human biomass movement. Low-income countries host 9% of the global population and only ~4% of human biomass movement. Before the late Pleistocene

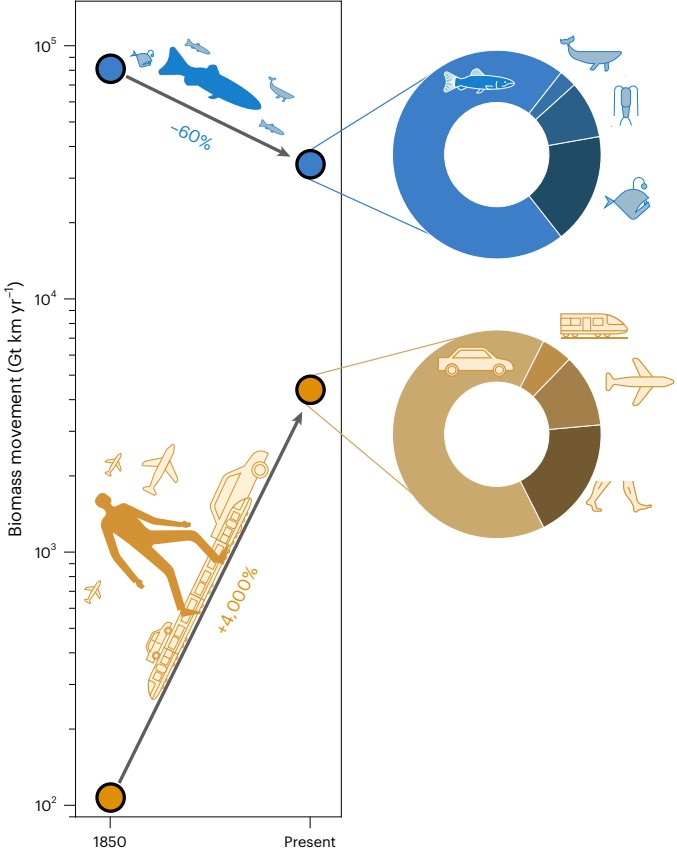

**Fig. 3 | Change in marine and human biomass movements and their compositions.** Marine biomass movement (blue dots) has decreased by ~60% since 1850 due to fishing, while human biomass movement (orange dots) has increased 40-fold. Most of the current aquatic biomass movement is due to the locomotion of pelagic fish (~70%), with mesopelagic fish (~15%), zooplankton (~10%) and mammals (~5%) contributing the rest—see upper doughnut chart. Most human biomass movement is currently in motorized road vehicles (~65%), followed by walking and cycling (~20%), flying (~10%) and rail transport (~5%)—see lower doughnut chart. See Supplementary Information for detailed estimates.

extinction (~50,000 years ago), before early human-driven extinctions, wild mammals had about ten times more biomass than today, mostly in megafauna[14] (body mass >44 kg, that is >100 lb). We estimate that their biomass movement was on par with current human walking and cycling (Supplementary Information).

**Table 1 | Energy used for locomotion by different animal groups**

| | Biomass movement (Gt km yr⁻¹) | COT average (J kg⁻¹ m⁻¹) | Energy per year (TWh yr⁻¹) | Power average (GW) | Human-associated power analogue |
|---|---|---|---|---|---|
| Wild land mammals | 30 | 3 | 20 | 2 | A large power station |
| Wild birds | 30 | 9 | 80 | 9 | Fleet of a major airline carrier |
| Marine mammals | 1,000 | 0.8 | 200 | 20 | All ships transporting natural gas and chemicals |
| Fish | 30,000 | 4 | 30,000 | 4,000 | All of human transportation |
| Humans walking | 600 | 3 | 500 | 50 | One-tenth of global caloric intake by humans |
| Humans, all transport modes | 4,000 | 30 | 30,000 | 4,000 | One-third of humanity's final energy use |

Estimates are based on the biomass movement of animals and allometric relations for the COT[19] and have an approximately threefold uncertainty (Supplementary Information). Bird energy use assumes active movement and the actual energy required might be smaller.

## Non-animal mass movement

Bacteria and other prokaryotes are collectively the most massive actively moving taxonomic group, with a global biomass of ~200 Gt (ref. [15]). Their average velocity is commonly much less than ~10 μm s⁻¹, so their active biomass movement is much less than ~50 Gt km yr⁻¹. Thus, animal biomass movement approximates all active living biomass movement. Human transportation uses ~1.3 billion cars, with a combined mass of ~2 Gt. We find that the mass movement of passenger vehicles amounts to ~40,000 Gt km yr⁻¹, similar to the biomass movement of all life on Earth (Fig. 2). Food transport accounts for most human-associated biomass movement. Such 'food miles' amount to ~8,000 Gt km yr⁻¹ (ref. [16]) or about twice the biomass movement of humans themselves. This nutrient transport is much larger than that facilitated by animals. Per person, food is transported ~3,000 kg km d⁻¹, primarily by ships[16]. Total cargo mass movement is still much larger, with international maritime trade transporting over 100,000 Gt km yr⁻¹ of freight[17]. The mass movement of oil alone is similar to that of all the passenger vehicles it powers.

## Energetic cost of movement

We can translate biomass movement into the energy used to achieve this movement. The COT describes the energy required for an animal to move a unit of its body mass over a unit distance[18]. An adult person, for instance, requires a total of ~3 J⁻¹ m⁻¹ (~0.7 kcal kg⁻¹ km⁻¹) when comfortably walking[18,19]. The COT typically obeys allometric scaling laws based on the body mass of an animal and its locomotion mode[20]. We multiplied biomass movement by the corresponding COTs to find that wild terrestrial mammals use ~20 TWh yr⁻¹, or ~2 GW in total power, as shown in Table 1. This is similar to the energy capacity of a single large power station. The power used for the air mobility of birds is also far exceeded when compared with its human counterpart, summing to the power used by a single major airline carrier. The average COT of land mammals is similar to that of a person since efficient large animals (with lower COT) dominate mammalian biomass movement. In contrast, cars and other motorized vehicles are much heavier than the passengers they typically carry, thus requiring ~30 J kg⁻¹ m⁻¹ when considering only passenger mass. Therefore, the overall use of energy for human transportation (~30,000 TWh yr⁻¹; ref. [21]) surpasses that of land vertebrates on the order of 300-fold and that of marine mammals ~100-fold, as seen in Table 1.

## Discussion

The results described here give a globally comprehensive quantification of biomass movement across animals and humanity. Our estimates account for limited data availability, resulting in an uncertainty of the overall biomass movement of ~3-fold for wild land animals, ~5-fold for wild marine animals and ~1.3-fold for humans and their vehicles (Supplementary Information). It includes uncertainties in total animal biomass[7,11,12,22] and the extent of their movements. Our estimate for the global biomass movement of insects is more uncertain than

that of birds and mammals, highlighting a knowledge gap that future research should aim to close. Higher-resolution animal tracking could substantially increase some estimated values by more fully accounting for small-scale movements[6]. Similarly, human mobility data in the Global South and for walking and cycling generally remain scarce. We thus complement our estimates with upper-bound estimates that are robust to such systematic biases and are typically based on the activity time and characteristic velocities.

Animals spend much energy on locomotion[23], facilitating vital ecological processes[4,24]. They transport nutrients, energy and other organisms as they move. They defecate, forage and are preyed upon and they physically change their environments by compacting soils, mixing waters and more. While our metric does not directly measure any of these processes, it relates to many. For instance, the substantial biomass movement of large animals reflects their ecological importance in transporting nutrients[25]. We find that most natural biomass movement is composed of non-migratory animals, suggesting the need for attention to their movement ecology. Similarly, non-migratory birds constitute two-thirds of avian biomass movement, but even their most abundant species have very limited movement data. A standing challenge is quantifying the global extent of the change in animal movement patterns in response to human disturbances[10,26–30]. The systematic survey presented here can help tackle these challenges by focusing future research on global mobility elements that are quantitatively dominant. Such research could also help to monitor biomass movement trends across scales and predict ecological outcomes such as nutrient cycling and energy expenditure.

While wild land animals (including invertebrates) outweigh humans roughly ten times in biomass[12], the systematic synthesis presented here reveals that human biomass movement exceeds the combined biomass movement of all terrestrial wildlife by an order of magnitude (Fig. 1). Large animals, which dominate the natural biomass movement, have declined on land and in the oceans over the past two centuries[25,30–32], as seen in Fig. 3. In contrast, human biomass movement has soared 40-fold (Fig. 3) due to population growth and utilization of motorized vehicles, fossil fuels and extensive mobility infrastructure systems. Today, a large power station generates as much power as is used for locomotion by all wild land mammals combined (Table 1). Similarly, iconic mass animal migrations pale compared with everyday human commutes (Fig. 2). While wild animals engage in remarkable migrations[2], the biomass movement of humans walking probably exceeds that of all terrestrial wildlife. Our findings offer another perspective to assess the current state of the biosphere. It complements other metrics[33–36] of environmental relevance in quantifying the magnitude of human actions compared with those of animals. Quantifying biomass movement across species and its associated energy costs provides a deeper understanding of the relationships between humanity and other species. It improves our perception and opens up new research avenues regarding their

interactions, trade-offs and potential pathways towards environmental sustainability.

## Methods

We evaluate total mobility as global biomass movement, which is the product of the global biomass of any given species and the total average distance its individuals travel within a typical year.

$$\text{Global biomass movement} = \text{global biomass} \times \text{total distance per year} \tag{1}$$

Each group required a distinct analysis due to variations in data availability and unique characteristics.

### Animal biomass movement

The total biomass for the various taxonomic groups was mostly taken from the literature (for example, ref. 12). The distance travelled was typically evaluated as an annual average, including most migrations and other movements. These data were evaluated on the basis of previous measurements and estimates, tracking data, models, or typical movement parameters such as speed and activity durations. For birds, we estimated such distances by original analysis using publicly available raw tracking data for ~6,000 individual birds. When aggregating taxonomic groups, such as all terrestrial mammals or all wild birds, the total biomass movement was first estimated on a single species level or for each subgroup with similar locomotion characteristics. The total biomass movement of the aggregated group was calculated as the sum of the total biomass movement of each species or each subgroup. When possible, we have made several independent biomass movement estimates as a consistency check and as a way to assess and mitigate possible errors. We assess the uncertainties of our estimates on the basis of the uncertainties in the underlying data and assumptions. A detailed description of all our biomass movement estimates is given in Supplementary Information.

### Human biomass movement

The biomass movement of humans was estimated in two steps. We first merged multiple global databases, country-level studies and scientific literature, using informed assumptions and our estimations to fill data gaps and harmonize the data. We derived estimates from the data for each mode of transport (motorized road vehicles, walking and cycling, flying and rail-based transport), considering missing information, population size and generalized assumptions. We developed estimates for all countries grouped into four income groups, as established by the World Bank[9], from which we also sourced population estimates. In the second step, we used two modelling approaches to account for human mobility where data were missing. The first model extrapolates the average biomass movement per person to the unreported part of the population of each income group. The second approach uses a regression model between each transportation mode and national GDP per capita to predict the average distances travelled in each country. The results presented are the sum of the data-driven estimates of the first step and the average of the two extrapolation approaches of the second step (Supplementary Information). Data availability and quality generally decline rapidly for lower-middle-income and low-income countries and for walking and cycling across most countries. We developed upper, mean and lower estimates to address the uncertainties in the underlying data, our assumptions and extrapolations.

We have used JupyterLab v.4.0.11 with Python v.3.12.4 and Microsoft Excel v.16.77.1 to analyse all the data.

### Reporting summary

Further information on research design is available in the Nature Portfolio Reporting Summary linked to this article.

## Data availability

The data generated and used to produce the results described in this study are available via Zenodo at https://doi.org/10.5281/zenodo.16731770 (ref. 37). The raw data of bird movement are available at https://www.movebank.org and can be downloaded using the code we provide.

## Code availability

The code used for data analysis, along with the code for downloading the raw data of bird movement, are available via Zenodo at https://doi.org/10.5281/zenodo.16731770 (ref. 37).

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

## Acknowledgements

We are grateful to the openness of many researchers who make their data publicly available and this research possible, including C. Carbone et al. from the Zoological Society of London, I. A. Hatton et al. from McGill University, the Movebank database and data owners, the AVONET database, BirdLife International, the IUCN and the referenced agencies, governments and organizations. We thank T. Antman, J. Belmaker, R. Ben-Nissan, R. Chen, Y. Cristal, G. Eshel, E. Galbraith, P. Henriksson, F. Krausmann, E. Lerchbaum, S. Lovat, U. Moran, R. Nathan, E. Noor, I. Raveh, U. Roll, K. Safi, M. Shachak and Y. Yovel for their help and valuable feedback during this study. We acknowledge funding from Schwartz-Reisman Collaborative Science Program (R.M.); Mary and Tom Beck-Canadian Center for Alternative Energy Research (R.M.); Institute for Environmental Sustainability at the Weizmann Institute of Science (R.M.); European Research Council through the European Union Horizon 2020 research and innovation program (MAT_STOCKS) grant no. 741950 (D.W. and D.V.). R.M. is Head of the Institute for Environmental Sustainability and an incumbent of the Charles and Louise Gartner Professorial Chair.

## Author contributions

Conceptualization: Y.R., D.W., L.G., L.A., R.P., R.M. Methodology: Y.R., D.W., D.V., R.M. Software: Y.R., G.B.-S. Validation: Y.R., D.W., D.V., G.B.-S., L.A. Formal analysis: Y.R., D.W., D.V., G.B.-S., L.G., B.H. Investigation: Y.R., D.W., D.V., L.G., B.H. Data curation: Y.R., D.W., D.V., G.B.-S. Writing—original draft: Y.R., D.W., D.V., R.M. Writing—review and editing: Y.R., D.W., D.V., R.P., R.M. Visualization: Y.R., D.W., R.M. Supervision: Y.R., D.W., R.M. Project administration: D.W., R.M.

## Competing interests

The authors declare no competing interests.

## Additional information

**Correspondence and requests for materials** should be addressed to Ron Milo.

# Reporting Summary

## Statistics

For all statistical analyses, confirm that the following items are present in the figure legend, table legend, main text, or Methods section.

| n/a | Confirmed | |
|---|---|---|
| ☐ | ☒ | The exact sample size ($n$) for each experimental group/condition, given as a discrete number and unit of measurement |
| ☒ | ☐ | A statement on whether measurements were taken from distinct samples or whether the same sample was measured repeatedly |
| ☒ | ☐ | The statistical test(s) used AND whether they are one- or two-sided<br>*Only common tests should be described solely by name; describe more complex techniques in the Methods section.* |
| ☒ | ☐ | A description of all covariates tested |
| ☐ | ☒ | A description of any assumptions or corrections, such as tests of normality and adjustment for multiple comparisons |
| ☐ | ☒ | A full description of the statistical parameters including central tendency (e.g. means) or other basic estimates (e.g. regression coefficient) AND variation (e.g. standard deviation) or associated estimates of uncertainty (e.g. confidence intervals) |
| ☒ | ☐ | For null hypothesis testing, the test statistic (e.g. $F$, $t$, $r$) with confidence intervals, effect sizes, degrees of freedom and $P$ value noted<br>*Give P values as exact values whenever suitable.* |
| ☒ | ☐ | For Bayesian analysis, information on the choice of priors and Markov chain Monte Carlo settings |
| ☒ | ☐ | For hierarchical and complex designs, identification of the appropriate level for tests and full reporting of outcomes |
| ☒ | ☐ | Estimates of effect sizes (e.g. Cohen's $d$, Pearson's $r$), indicating how they were calculated |

*Our web collection on statistics for biologists contains articles on many of the points above.*

## Software and code

Policy information about availability of computer code

Data collection | *Provide a description of all commercial, open source and custom code used to collect the data in this study, specifying the version used OR state that no software was used.*

Data analysis | We have used JupyterLab version 4.0.11 with Python version 3.12.4 , and Microsoft Excel version 16.77.1 to analyze data. The data and code are publicly available at https://doi.org/10.5281/zenodo.16731771

For manuscripts utilizing custom algorithms or software that are central to the research but not yet described in published literature, software must be made available to editors and reviewers. We strongly encourage code deposition in a community repository (e.g. GitHub). See the Nature Portfolio guidelines for submitting code & software for further information.

## Data

Policy information about availability of data

All manuscripts must include a data availability statement. This statement should provide the following information, where applicable:

- Accession codes, unique identifiers, or web links for publicly available datasets
- A description of any restrictions on data availability
- For clinical datasets or third party data, please ensure that the statement adheres to our policy

All data generated and used to produce the results described in this study is publicly available at https://doi.org/10.5281/zenodo.16731771.
We have also used data from Movebank database which can be downloaded via its API with the provided code.

# Research involving human participants, their data, or biological material

Policy information about studies with <u>human participants or human data</u>. See also policy information about <u>sex, gender (identity/presentation), and sexual orientation</u> and <u>race, ethnicity and racism</u>.

| | |
|---|---|
| Reporting on sex and gender | The data on human mobility was collected by various organizations independently of this work, as described in the main text and supplementary information. It typically did not include information regarding sex or gender. Our results regarding human biomass movement apply to both sexes. |
| Reporting on race, ethnicity, or other socially relevant groupings | We have used the grouping of countries into standard economic income groups, as developed by the World Bank. This categorization was part of our biomass movement extrapolation procedure, as countries under the same economic income group share relevant human mobility characteristics. Further details are found in the supplementary information. |
| Population characteristics | *Describe the covariate-relevant population characteristics of the human research participants (e.g. age, genotypic information, past and current diagnosis and treatment categories). If you filled out the behavioural & social sciences study design questions and have nothing to add here, write "See above."* |
| Recruitment | *Describe how participants were recruited. Outline any potential self-selection bias or other biases that may be present and how these are likely to impact results.* |
| Ethics oversight | *Identify the organization(s) that approved the study protocol.* |

Note that full information on the approval of the study protocol must also be provided in the manuscript.

# Field-specific reporting

Please select the one below that is the best fit for your research. If you are not sure, read the appropriate sections before making your selection.

☐ Life sciences   ☐ Behavioural & social sciences   ☒ Ecological, evolutionary & environmental sciences

For a reference copy of the document with all sections, see nature.com/documents/nr-reporting-summary-flat.pdf

# Ecological, evolutionary & environmental sciences study design

All studies must disclose on these points even when the disclosure is negative.

| | |
|---|---|
| Study description | We synthesized hundreds of studies and data sources and utilized diverse approaches to coherently evaluate the biomass movement of all animals and humans on Earth. |
| Research sample | Our data is based on the published literature, the Movebank database, the AVONET database, BirdLife International, the IUCN, and the referenced agencies, governments, and organizations. It includes information on animal biomass, mobility, and other characteristics. |
| Sampling strategy | We used all the relevant data that we could find to form our samples. |
| Data collection | Data was collected from published literature and databases by YR, DW, DV, GBS, LG, BH. We did not collect original data (e.g. we did not track animals ourselves) |
| Timing and spatial scale | Our data spans across the world and multiple years, seasons, and times of the day. We aimed at collecting data that can be averaged on an annual basis. |
| Data exclusions | We excluded bird movement data we suspected to be erroneous or could not be used to infer typical daily movement patterns, as detailed in the supplementary information. The exclusion criteria were pre-established. |
| Reproducibility | Our study does not include experiments. |
| Randomization | We grouped our animal data by taxonomy and trait and tested the sensitivity of the results to the various groupings, as discussed in the manuscript. The effects of the related potential biases is incorporated into our uncertainty analysis. We also estimate upper bounds as an alternative way to account for potential biases in the data and data scarcity. |
| Blinding | Blinding was irrelevant to our study during data acquisition, as the data was preexisting, objective, and we collected all the available data we could find. Blinding of the animal taxonomy was not possible during analysis, as the analysis was partly based on taxonomy. However, the analysis was based on objective measures and established protocols when possible, mitigating potential biases. |

Did the study involve field work?   ☐ Yes   ☒ No

# Reporting for specific materials, systems and methods

We require information from authors about some types of materials, experimental systems and methods used in many studies. Here, indicate whether each material, system or method listed is relevant to your study. If you are not sure if a list item applies to your research, read the appropriate section before selecting a response.

## Materials & experimental systems

| n/a | Involved in the study |
|-----|----------------------|
| ☒ ☐ | Antibodies |
| ☒ ☐ | Eukaryotic cell lines |
| ☒ ☐ | Palaeontology and archaeology |
| ☒ ☐ | Animals and other organisms |
| ☒ ☐ | Clinical data |
| ☒ ☐ | Dual use research of concern |
| ☒ ☐ | Plants |

## Methods

| n/a | Involved in the study |
|-----|----------------------|
| ☒ ☐ | ChIP-seq |
| ☒ ☐ | Flow cytometry |
| ☒ ☐ | MRI-based neuroimaging |

## Plants

| | |
|---|---|
| Seed stocks | *Report on the source of all seed stocks or other plant material used. If applicable, state the seed stock centre and catalogue number. If plant specimens were collected from the field, describe the collection location, date and sampling procedures.* |
| Novel plant genotypes | *Describe the methods by which all novel plant genotypes were produced. This includes those generated by transgenic approaches, gene editing, chemical/radiation-based mutagenesis and hybridization. For transgenic lines, describe the transformation method, the number of independent lines analyzed and the generation upon which experiments were performed. For gene-edited lines, describe the editor used, the endogenous sequence targeted for editing, the targeting guide RNA sequence (if applicable) and how the editor was applied.* |
| Authentication | *Describe any authentication procedures for each seed stock used or novel genotype generated. Describe any experiments used to assess the effect of a mutation and, where applicable, how potential secondary effects (e.g. second site T-DNA insertions, mosiacism, off-target gene editing) were examined.* |

