## [Peer Review File · Nature Ecology & Evolution]

Human biomass movement exceeds the biomass movement of all land animals combined

Corresponding Author: Professor Ron Milo

This manuscript has been previously reviewed at another journal. This document only contains information relating to versions considered at Nature Ecology & Evolution.

Version 0:

Decision Letter:

9th June 2025

Dear Dr. Milo,

Thank you for submitting your revised manuscript "Human biomass movement exceeds the biomass movement of all land animals combined" (NATECOLEVOL-25041007-T). It has now been seen again by two of the original reviewers and their comments are below. The reviewers find that the paper has improved in revision, and therefore we'll be happy in principle to publish it in Nature Ecology & Evolution, pending minor revisions to satisfy the reviewers' final requests and to comply with our editorial and formatting guidelines.

****Please note**** We are not able to accommodate References that are cited as 'In prep'. Thank you for your explanation about the relationship between this manuscript and Reference 9 "The Global Biomass of Mammals Since 1850"; having discussed this further with the editorial team, we feel that there are two options on how to proceed: 1) upload this 'in prep' manuscript as a preprint so that it is citable, and provides transparency on any methods that relate to it; or 2) remove this reference from the manuscript and re-analyse those estimates that cite it. If you choose the latter option, we will have to seek an additional review round with one or more of the previous reviewers in order to verify that their assessments still stand.

If you have not done so already, please ensure that you also email us completed copies of the Reporting summary and Editorial policy checklists:

Reporting summary: https://www.nature.com/documents/nr-reporting-summary.pdf
Editorial policy checklist: https://www.nature.com/documents/nr-editorial-policy-checklist.pdf

Sincerely,

[redacted]

<https://www.nature.com/natecoevol/>
@natureecoevo.bsky.social

Reviewer #1 (Remarks to the Author):

I have read through the responses to reviewers and again feel the authors have done a good job responding to the reviewer's comments. They have fully addressed my previous concerns and greatly clarified the section regarding the biomass movement of the Pleistocene era. It is difficult to do full error estimates when data is lacking, but they have done a good job under such circumstances.

Reviewer #2 (Remarks to the Author):

I have now had a chance to review the revised version of the manuscript by Milo et al., and the authors have addressed all of my comments. While we have differing views on some points, which are largely matters of interpretation or emphasis, I do not see any issues warranting further revision. I appreciate the authors' efforts in revising the manuscript and engaging with the feedback and am therefore happy to recommend acceptance of the manuscript in its current form.
